# Enriching Athlete—Environment Interactions in Youth Sport: The Role of a Department of Methodology

**DOI:** 10.3390/children10040752

**Published:** 2023-04-20

**Authors:** Keith Davids, Martyn Rothwell, Sam Hydes, Tim Robinson, Charlie Davids

**Affiliations:** 1Sport and Physical Activity Research Centre, Sheffield Hallam University, Collegiate Hall, Collegiate Crescent, Sheffield S10 2BP, UK; 2Department of Sport, Exercise and Nutrition Sciences, School of Allied Health, Human Services & Sport, La Trobe University, Melbourne, VIC 3086, Australia

**Keywords:** ecological dynamics, coaching, department of methodology, transdisciplinarity, skill adaptation, ecological enrichment, affordances

## Abstract

The aim of this insights paper is to propose how the theory of ecological dynamics may invite re-consideration of how sport scientists could support performance, learning and development of children and youth in sports programmes. We seek to outline why learning should be *individualised* and *contextualised*, based on the specific needs of learners, such as children and youth, women and disabled athletes in sport. Case examples from individual and team sports are presented to illustrate how constraints can be designed to enrich interactions of children and youth with different performance environments, based on integrating principles of specificity and generality in learning and development. These case examples suggest how a collaborative effort by sport scientists and coaches in children and youth sport may be undertaken in a department of methodology to enrich learning and performance.

## 1. Introduction

Re-imagining sport science education and training to support development of individuals in specific sub-groups in sport is necessary because of a historical tendency to use data from studies of adult male athletes to plan programmes for athlete development and performance preparation for universal implementation. This re-consideration is needed in children and youth sport programmes because there has been a historical tendency to inappropriately treat them as “miniature adults”. For example, in the socio-historical culture of colonial Puritan communities in North America, the needs and desires of children and youth were not recognised, nor were specific societal roles and identities specified for them [1]. According to Beales [1], this culturally entrenched view on children’s place in society also impacted on common psycho-social attitudes to child-rearing practices.

More recently, Rothwell et al. [2] documented how these socio-cultural and historical constraints have continued to shape childhood and youth development experiences in formal sport programmes in more recent times. They identified the increasing professionalisation and “adultification” of junior sports programmes as a significant threat to the health and wellbeing of children and youth in sports participation programmes at different levels of skill and expertise [3].

Related to this issue, a similar argument can be made with regards to the challenge of working with other groups in sport, including women and paralympic athletes. Indeed, this insights paper posits the need for theoretically based contextualised training programmes for learning and development of individuals in specific sub-groups in sport programmes, exemplified by, but not limited to, children and youth, women and groups with disabilities. Individuals in all groups have their own specific psycho-physiological needs framed by socio-cultural and historical expectations.

### 1.1. Children and Youth Are Not ‘Mini-Adults’

A key principle of an ecological dynamics-oriented application of sports sciences to facilitate developmental experiences in competitive sports is that children (and others) should not be treated as “honorary male adults” during training and practice for the purposes of performance preparation and athlete development. Not treating children as “mini-adults” places the individual child or young person at the centre of learning and development processes with which they are challenged throughout the early span of the life course [4].

The aim of this insights paper is to review how training designs in sport programmes can be based upon general principles to provide relevant sports science and sport pedagogical support for all groups. Here, we argue that *contextualisation* of practice is vital, based on the specific needs of different athlete populations, as well as *individualisation* of practice and training designs [3]. To support this rationale, we critically reviewed the existing literature in ecological dynamics, and papers from sports science research relevant to examining the argument that the needs of different sub-groups (e.g., children, women and disabled individuals) have been under-investigated due to the dominant tendency to conduct studies on male, adult athletes. More specifically, this critical analysis allowed us to unpack what contextualisation and individualisation means in terms of sport science support for youth and children. To achieve our aim, we conducted an overview of the existing literature, providing a critical analysis as to why an ecological dynamics rationale can explain why contextualised and individualised sport science support is vital for sub-groups of participants at all performance levels, such as children and youth. Three case examples from individual, team sports and training are provided to bring to life the theoretical ideas explored in our critical review.

### 1.2. An Ecological Dynamics Rationale for Development of Youth in Sport

Some calls to avoid treating children as adults in sports training have been evidence-based, for example from a health and safety perspective [5,6]. Some sport development models, with a practical application focus, such as the long-term athlete development model [7,8] and youth physical development model [9,10], consider the maturational status of the child [as opposed to simply relying on chronological age], offering a strategic approach to athletic development of children and youth. However, these models have not provided a detailed theoretical explanation for why the “adultification” of children’s experiences in sport needs to be avoided, focusing on practical implications (e.g., injury repercussions) instead. Furthermore, individual development in such models is often considered in isolation, for practical purposes, separating physical, technical, tactical and psychological skills training [11]. The assumption is that the performance of a young athlete can be improved by simply addressing a single component of a larger, complex system at a time.

In contrast, the contemporary theoretical framework of ecological dynamics provides a complementary integration of concepts, ideas and tools from nonlinear dynamics and ecological psychology to contextualise and individualise each child’s experiences in sport development [3,12]. Key ideas in ecological dynamics have been applied to the study of sport performance in individual athletes and teams, addressing motor learning, the acquisition of expertise and talent development throughout the past few decades [13]. The principles of ecological dynamics can help with the challenge of enriching the learning experiences of children and youth participating in sports programmes at all skill levels to facilitate implicit skill adaptation. Self-organising actions implicitly, to satisfy interacting constraints in sport, supports performers (conceptualised as nonlinear dynamical systems) to perceive surrounding information (e.g., visual, proprioceptive, haptic and acoustic) to regulate intentions and use cognition (for problem-solving to make choices and decisions) [14,15].

Nonlinearity in dynamical movement systems refers to the varied rates at which different sub-systems of each individual change over time, through growth, maturation and development. This important idea captures how children, youth and adults are continuously changing, at different timescales (e.g., learning and development), throughout the life course under the influence of various interacting constraints [16]. Nonlinearity of development and maturation is a significant challenge for sport scientists engaged in identification, prediction and assessment of “talent” at an early age [17]. For this reason, identification, selection and development of young children in early specialisation programmes in sport have been rejected as ineffective and inefficient uses of time and resources in sport science (see Coutinho et al., this special issue). Instead, sport scientists should improve their focus on *enriching the interactions* of individuals with sport environments to provide the most effective and efficient opportunities and experiences for learning and development, rather than allocate resources and efforts on identification and selection of individuals at an early age, relative to each sport’s norms [18].

### 1.3. Newell’s Constraints Model

Newell’s [19] model of interacting constraints implies that a most effective way for sport scientists, coaches and teachers to guide performance, learning and development in sport is to manipulate individual, environmental and task constraints continuously guide the development of each individual in practice.

*Individual constraints* include characteristics and properties of movement systems at a specific phase of maturation and development in childhood and adolescence [16]. Different sub-systems of each individual change at different rates across varying timescales. This important point presents individual children with challenges during performance, learning and development. The sub-systems of each child include those which guide action, perception, cognition, emotions, hormones production and more throughout the life course [16]. Each sub-system may change at different rates in each individual, presenting a strong case for individualised sport science support of youth athletes by groups of specialist practitioners working together in developing practice and training programmes that are relatively unique. Even within the same age group (e.g., athletes aged 13–14 yrs) there will be a significant amount of variability with regards to maturation and development of each sub-system, as well as different levels of learning, education and experience. For this reason, nonlinearity of development, encompassed by inter-individual and intra-individual variability in action capabilities and performance capacities, should be considered to be typical throughout childhood [3,13]. Treating every child in the same way will not provide equality of opportunity for children and youth to achieve their potential [3,20]. Consequently, *individualisation* of development and preparation for performance should be based on the specific personal constraints of each child in the design of youth sports programmes for learning and development.

*Task constraints* refer to the conditions of practice design, e.g., practice area dimensions and locations, intended aims of the task, equipment and technology used to enrich interactions of children and the learning environment [13]. An ecological rationale for technology use for learning and development of children proposes that sport scientists should not seek to replace the children’s direct experience with sport contexts [21]. The implication is that teachers and coaches should limit the amount of indirect experience that children and youth have (e.g., watching video films of athletes performing skills instead of engaging directly with sports performance environments to gain primary experiences). These ideas have important implications for how coaching practitioners and sport scientists use, and their attitudes towards, technologies which contribute to the secondary experiences of young people during learning. Even when educators and practitioners intend to provide primary learning experiences, technology often becomes the main contributor to designing practice landscapes. Technology-based learning contexts are less likely to provide *direct* experience of competition such as primary encounters in practice and performance. In physical education and sports, an important aim for sport scientists is to consider how technology can enrich interactions between athletes, coaches, content (e.g., areas of performance to be explored), and a learning and development context. Essentially, technology can be used to strengthen and support reciprocal interactions within the athlete–coach–environment learning system. Too often, technology gets in the way of these interactions, because it is believed to provide what practitioners consider to be “an objective truth” [e.g., statistical data] about what performance issues might need “fixing” in a team or an individual athlete. The over-use of statistical performance data in coaching youth can encourage explicit learning methods, rather than facilitate implicit skill adaptation and, of course, there is much more to sport performance that sport scientists need to consider.

A major task constraint under the remit of sport scientists includes designing *affordances* [22,23] in the practice landscape. According to Gibson [22], affordances are not causes of a movement, but rather are *possibilities or opportunities for action* which people may or may not use to achieve their intended performance outcomes. In ecological dynamics, practice is a search process for children to find and use available affordances as opportunities for action in the sport performance landscape [24]. This important ecological idea implies that the environment is a “manifold of action possibilities” [23] with which a young person needs to learn to interact in sport practice.

A sport science team’s role is to guide these interactions by helping young athletes to learn to attend to available affordances in a performance environment: to seek, discover and exploit them to enhance their performance functionality during learning. According to Jacobs and Michaels [25], one can guide an athlete’s intentions in negotiating a performance environment, with regards to specific aims, objectives and performance goals which may support decision making and problem solving in youth sport.

These ideas of Withagen et al. [23] suggest that task constraints for motor learning in sport cannot improve utilisation of affordances in young learners by making them more prominent (e.g., providing bigger gaps between defenders for attackers to pass through in a team games drill). Rather, affordance utilisation in youth sport can best be improved by better aligning the effectivities or action capabilities of individual children with the information and the affordances that become available in a performance environment. In ecological dynamics, a performer’s effectivities refer to their personal capacities, predispositions, underlying abilities and tendencies for performance which allow them to use a range of affordances for action that emerge in play, recreation and sport. To achieve this purpose, task constraints should be designed to be more *neutral* [23] to support the improved functionality of the athlete–environment relationship. This idea aligns neatly with Bernstein’s [26] conception of practice as ‘repetition without repetition’ [repeatedly seeking different performance solutions in practice], a key principle in ecological dynamics.

In contrast, a popular principle of design in everyday life is that many objects and locations in life can have *limited* purposes [23]. It is the same in team sports and other dynamic performance environments. In junior volleyball, for example, drills are often designed to deliberately manipulate practice task constraints for athletes to achieve a specific performance outcome. This type of prescriptive task constraint design is common in many sports where *repetition and “choreography” of a movement technique* is over-emphasised e.g., simply rehearsing an “ideal” technique for smashing a volleyball in attack, without opposition present at the net, and without consequences if the ball is hit out of court [27]. This “technique rehearsal” approach overlooks the search for solutions during practice to problems of avoiding a two- or three-person opposition block, for example. Using a “technique repetition” approach, coaches typically prescribe few opportunities for decision making or problem solving, including a lot of pre-determined movement sequences with which athletes need to comply.

Instead, by designing more *neutral* task constraints, sport scientists and coaches can facilitate adaptability, creativity and innovation in children. To exemplify, in volleyball, coaches can encourage children to explore neutral performance landscapes by adding defenders and changing attack location at the net and ensuring there are consequences if the ball is hit into the net or out of court by the attackers. Games and play activities with *Neutral* task constraints in sports coaching and teaching have many affordances, appearing and disappearing, depending on the ebb and flow in the dynamics of a well-designed “open” practice environment [11,28]. In such a sport practice landscape, the aim is to encourage children to search, discover and explore affordances which are more diverse [3]. Designing practice task constraints which are more *neutral* in terms of performance outcomes could better simulate the constraints of the competitive performance environment, encouraging children to problem solve, make decisions and learn to use different affordances during practice. With this coaching methodology learning designs in sport practice could resemble “repetition without repetition”, including *solicitations* for functional actions in a specific field of promoted actions [e.g., in a transitional phase of play in basketball] to specify the learning context of an individual child or team during practice [3].

*Environmental constraints* include socio-cultural and historical factors that can manifest in strong traditions and attitudes towards developmental practices, such as taken-for-granted misconceptions for treating children and youth as mini-adults in society [1]. In children’s sport programmes, socio-cultural-historical constraints can act as adverse *rate limiters* on children’s development, but are not always tangible, and in many cases serve to hinder children’s development, unbeknown to parents and practitioners. To fully embrace established principles of child development in ecological dynamics, from the perspective of working with children as a special population, we must not only consider factors within the child’s immediate environment [e.g., physical, geographic and developmental experiences], but also their interactions within the wider environment [e.g., social and cultural systems and historical tendencies].

To understand the wider ecology of a child’s development, the prominent ecological psychologist Urie Bronfenbrenner [29,30] proposed a bioecological theory of human development. A key feature of Bronfenbrenner’s work is the process–person–context–time model [31]. Bronfenbrenner’s ecological conceptualisation highlights the interrelatedness, and proximal processes between individuals and the (immediate and wider) environment within which they are situated, as important drivers of their development. In essence, development is an emergent property emanating from the synergistic and reciprocal interactions between individuals and their environments. Applying this idea to sport development in children, environments are considered to be complex and interconnected systems (i.e., macro—e.g., cultural norms, exo—e.g., the influence of sport scientists on a performance pathway, meso—e.g., school/education and micro—e.g., day-to-day practice designs), that shape (positively or negatively) an aspiring athlete’s developmental experiences. Take, for example, the unacceptable culture of British Gymnastics World Class Programmes, a culture that was born out of an unhealthy obsession with competitive success (macro level), leading to performance preparation practices that normalised the physical and emotional abuse of young gymnasts during training (micro level) [32]. These issues were confounded by poor safeguarding practices and procedures within British Gymnastics clubs (exo-level), meaning that many elite level gymnasts, and parents, felt unable to raise their concerns with the relevant authorities (meso-level) [32].

### 1.4. Enrichment and Task Constraints with Neutral Affordances

In sports organisations, sport scientists could advise coaches and teachers on how to design task constraints and skill adaptation programmes that enrich the individual-performance environment relationship, providing greater functionality in children and youth. Enhanced functionality needed for successful interactions with a performance environment can emerge through enriching interactions: at specific and general levels [3]. Coaching frameworks such as nonlinear pedagogy and the athletic skills model (ASM) advocate that children and youth should undertake a complementary programme of less-specialised (general activities involving play and practice in multiple sports) as well as more-specialised (specific to one or two sports) movement experiences prior to advanced training. A nuanced balance is needed between *generality* and *specificity* of practice during learning in childhood and youth phases of sport development. These ideas align with those of Anatoly Bondarchuk [33], a sport scientist who proposed a complementary relationship between generality and specificity of training for young athletes on sport development pathways. This complementarity provides the basis for developing athleticism and foundational movement capacities needed to make the most of specialised training in a single sport at a later phase of development.

Specific aspects of a play or practice environment could include the property of *“inviting potential”* [24], predisposing children to seek and explore certain performance tendencies as functional outcomes. Sport scientists could design soliciting effects in practice which invite individual athletes or teams to perform in a relational way with the affordances of a performance environment, helping them to learn to accept or resist their influence. This approach to practice provides the basis for athlete self-regulation in sport, explaining autonomy and agency at the performer–environment scale of analysis. According to Reed [34] self-regulation is predicated on the idea that “organisms make their way in the world”. This is a useful way to consider how children and youth could be encouraged to negotiate the trials and tribulations of competitive performance.

For example, this idea has important implications for the design of athlete development pathways, such as the elite pathway for players in team sports such as rugby union, which comes in to play at under 14 yrs of age. “Talent” selection in rugby union is traditionally undertaken through implementing reductionist tests of anthropometric measurements, technique assessments in isolation and small-sided games, with a child’s performance scored through a tick sheet composed by the elite club involved. In the UK, judgements about “talent” are made by the localised coaches working in each Developing Player Programme (DPP). An elite club works with affiliated junior clubs and this does have an impact on the volunteer coaches and how they coach. They observe these procedures as part of the coaching system that they need to affiliate with and aspire to. Although the Rugby Football Union in the UK views rugby as a late specialisation sport, selection is undertaken at age 13 and 14 yrs, which brings significant challenges for the player. This idea also ties in well with key ideas of Bronfenbrenner’s model [discussed earlier], since the players go to the DPP once a week, but then play and train with other local club teams and in school, where they are faced with different socio-cultural values and traditions.

### 1.5. Individual Constraints Can Be Rate Limiters

Sub-systems of an individual’s movement system, developing at different rates, can act as rate limiters on performance and functionality in some children at varied states of development. For example, sub-systems for regulating actions requiring dynamic power, agility, flexibility, strength and endurance may vary in development throughout adolescence in some individuals due to different rates of maturation and physical and hormonal changes. These developmental variations may result in performance decrements at critical phases of junior sport programmes [35]. This individual variation has implications for how sport scientists plan, organise and monitor performance in competition and training in youth sport programmes. The level of intra- and inter-individual variations in competition and training may vary more in youth athletes than in adult performers, which needs to be recognised in terms of planning the intensity, duration and the nature of recovery activities.

### 1.6. What Should Practice Look like in Youth Sport?

A key implication concerns repetition in training and practice. Endless repetition and rehearsal of skills and exercises is not recommended in performance preparation and development of youth athletes, due to potential physical and psychological impacts. Rather, Bernstein’s [26] conceptualisation of practice as “*repetition without repetition*” in the search for functional movement solutions implies an emphasis on the *quality* of practice, rather than quantity, as advocated by some other approaches to expertise, such as deliberate practice [36]. Bernstein [26] emphasis on “repetition without repetition” develops dexterity in athletes, essentially allowing an individual to form an adaptive relationship with a performance environment in individual and team sports. This has been termed “skill adaptation”, allowing children and youth to move in a dexterous and functional way in sport performance contexts [37]. In acquiring functional dexterity, play has been proposed as particularly important for young organisms of any species, including human children and youth. In children and youth, play is important in development of cognitive, perceptual and motor system functioning [15,16] and is important in early childhood learning experiences in schools for developing *physical literacy* [11,38].

### 1.7. Physical Literacy and the Athletic Skills Model: Enriching Interactions of Children in Sport Practice

Enrichment in childhood and youth phases of development facilitates skill adaptation as proposed in an ecological dynamics rationale for physical literacy [11]. These ideas, predicated on principles from the ASM, advocate the importance for sport scientists for engaging with first the child, then the mover and finally the athlete. A functional balance between generality and specificity of training and practice can enrich the interactions of an individual’s *effectivities* [22] and available affordances in different sport contexts. Sports training environments can be designed to be more soliciting of affordances which are closely related to the effectivities of each individual performer, facilitated by a transdisciplinary approach to performance, learning and development [39]. A transdisciplinary approach to the development of young athletes individualises and contextualises practice and training in sport. As we outline in more detail next, teams of sport scientists could work together in a department of methodology to design individualised development programmes based on the needs of each child/youth. These early sport experiences for children and youth are essential to enrich their relationships with sport performance environments, supporting them to explore and exploit the relations between their effectivities and available affordances. The relevant enrichment of interactions of children and youth athletes starts with the formal education environment at school age where sound habits could be developed as a result of formal experiences in physical education, as well as informal, unstructured play activities with peers and parents in the playground, parks and streets [11]. Interactions in family and peer group settings at the meso-level [29,30] can support the development of child and youth athletes in adopting important habits for an active lifestyle, related to nutrition, play and physical activity, education and study and rest.

### 1.8. Defining Enrichment of Interactions with the Environment: A Nuanced Balance between Specificity and Generality of Practice

This balanced and more nuanced approach to enrichment and development of children and youth in sport avoids the physical, psychological and emotional pitfalls of early specialisation in childhood and adolescence. The main role of enrichment activities is to support individual–environment interactions. The emphasis should not be on enriching the child/youth or the practice environment universally and separately. Enrichment should be scaled to the specific child/youth–environment *relationship* in sport. This subtle aim can be achieved through carefully designed, developmentally-appropriate practice and training tasks which engage individuals in solving problems, facing challenges and making decisions—all with functional actions as the outcome.

Therefore, a balance between general and specific practice and play activities is important for youth athletes in interacting with performance and practice environments (as advocated in athlete development models such as the ASM). An ecological dynamics rationale links dexterity, functional variability and skill adaptation to the integration of S&C with motor learning. The implication is that individual needs in children and youth can be best understood, planned for and met by coordinated work of sport science specialists working in a collaborative partnership, for example nutritionists, educators, S&C staff, skill acquisition specialists, psychologists, movement scientists, physiotherapists and the like. It has been shown how the work of collaborating specialists can be well organised in a department of methodology [28].

### 1.9. The Role of a Department of Methodology to Support Integrated Practice

A key implication of an ecological dynamics rationale is that the holistic development of children (e.g., skills work, tactical development, strength and conditioning (S&C) training and cognitive skills such as problem solving and decision making) should be integrated into specially designed games and practice activities by a group of collaborating sport scientists and coaches. Typically, distinct attributes for performance are viewed as isolated components that are developed independently from one another. This type of reductionist thinking can lead to further levels of separation, evident in so-called “skills training”, where specific techniques are developed in isolation (e.g., unopposed technique practice of passing, dribbling and heading in soccer). These approaches have tended to operate in a linear and rational way, where hierarchical staff structures limit integration and collaboration between other practitioners in the wider multidisciplinary team. Criticisms have been aimed at this linear and reductionist way of operating, with complexity science being proposed as a theoretical framework to update taken-for-granted methodological principles to guide integrated sport science support [40]

To move away from reductionist and siloed practices, researchers [28,41,42] have conceptualised a transdisciplinary framework for sport scientist and coach integration called a department of methodology (DoM). At the core of a DoM is a view of transdisciplinarity that (re)positions a collective of practitioners as an integrated, inquiry-based unit collaboratively solving development- and performance-based problems [41,42]. This repositioning strengthens the focus on enriching specific *child-environment interactions* in practice and training, by identifying the needs of each child, based on skill level, maturation and development, and the nature of their previous experiences. Through an inquiry-based approach, a holistic perspective encourages a stronger relationship between the inquirer [e.g., sport scientists, coaches and the children themselves] and the inquiry [what the immediate development needs of the child are] [41], so enrichment activities can be designed in full consideration of the multidimensional factors that contribute to a child’s learning and development.

Adopting a DoM in this way provides a framework to encourage a vibrant ecosystem that facilitates healthy and valuable knowledge transfer between key personal [including the child]. For example, team members with relevant knowledge and experience of coaching pedagogy, skill acquisition and performance analysis could work together to evolve practice experiences that rejects universal and decontextualised learning tasks. Collaboratively designing learning and development experiences that offer *neutral* practice landscapes rich in opportunities to explore, discover and exploit new action capabilities can more effectively strengthen child–environment interactions [3]. In a similar way, a developmental paediatrician, S&C coach, and technical and tactical coach could collaborate to identify appropriate physical demands, according to maturation levels, when designing small-sided games that stress an individual’s full range of skills. In summary, under the guidance of a DoM, applied scientists responsible for the growth of children can develop better interconnections to identify the dynamic properties that can enrich the child-environment system.

The concept of integrating generality and specificity of practice (linked to the ideas of Bernstein and of Bondarchuk), requires a better integration between subdiscipline specialists that can support more effective transitioning along a continuum of specificity and generality of practice, leading to refined interactions opportunities for enrichment. This leads to an important question for sport scientists: what type of practice does an individual child need/want and when? Based on these needs, manipulation of task constraints can support athlete learning and development and performance preparation by scaling performance area dimension, the numbers of athletes involved in practice designs, equipment and technologies used and rules and challenges presented to learners [43]. This analysis provides important information in responding to important questions for sport scientists working with children and youth in team games; for example: when are small-sided and conditioned games needed over full sided games? When might an individual sport athlete/child need to compete up or down an age group? What specific effects may specific small-sided and conditioned games have on children’s effectivities (e.g., 2v2, 4v4 and 7v7)?

This integrated approach differs from the conventional approach to performance development of children and youth in sport, which emphasises isolated training and practice of techniques, or physical training separately [18]. Reductionism in development can lead to an exaggerated over-emphasis on generality of practice, with sports-specific skills only being integrated in training later in the development pathway. Indeed, the development of functional movement patterns is necessary to promote physical literacy and create a movement foundation during childhood and youth phases [continuing in later adulthood]. It is often also necessary to utilise generality in practice to stress and overload physiological systems, thereby achieving physical adaptations such as improved strength or speed [33].

However, ecological dynamics proposes that greater specificity in practice tasks may also be introduced *alongside* such general skills training to simultaneously develop all sub-systems underpinning performance. To achieve this aim, fundamental motor skills can be blended into various games and play activities for children. Skill can be refined by varying, dynamic and challenging environments to enable children to use perception, cognition and action to become more capable of solving various movement problems they will encounter in their sport. In a department of methodology, a transdisciplinary approach incorporating better integration between subdiscipline specialists would facilitate more effective movement along the continuum of specificity and generality of practice, leading to better enrichment of interactions. In the following section, we discuss several practical case examples of how child–environment interactions may be enriched in individual- and team-sport skills training integrated with S&C training.

## 2. Practical Case Examples

### 2.1. Enrichment in Individual Sports (Diving)

Training environments for diving are based in dry-land contexts and the pool, with many tasks being decomposed into segments to facilitate performance improvement and development [44]. It is assumed that reducing a whole movement into isolated, manageable subcomponents will lead to successful performance of the entire task when parts are re-integrated together in competitive performance [45]. Coaches often devise drills that emphasise repetitive rehearsal of *perfect* movement techniques. This approach to practice is based on the traditional (false) assumption that, once a movement technique is well learned, stable and “programmed internally”, it is more resilient to the stresses of a dynamic competitive environment [20]. This traditional approach follows a historical progression in practice design of transitioning tasks from *simpler to more complex*, by deconstructing the complexities of a whole skill into parts. However, the risk is that the coach may disrupt the development of important information–movement couplings, thereby inhibiting skill adaptation [27]. Barris et al. [44] found that different coordination tendencies emerged (e.g., variations in jump height and hurdle-step length) when comparing the preparation phase of the dive in the dry land and pool. These coordination differences observed between the dry-land and pool environments were likely due to the change of task constraints encountered by the diver due to task decomposition. In contrast, a non-linear pedagogical approach advocates *task simplification*. Task simplification is a process of maintaining information–movement couplings and involves making tasks simpler without disrupting the information–movement couplings that regulate behaviour [13].

### 2.2. Performance Problem Exemplar: Diver Landing/Entering the Water Too Far Away or Too Close from the Board

A diver’s aim is to land a safe distance away from the diving board without diving too far into the pool. Diving too close or too far from the board (see Figure 1) will have an impact on their overall scores, facing a 0.5–2-point deduction from the judges [46]. Landing too far out will not only have an impact on their overall score but will change how they interact with the diving board, their rotational speed, the shape quality, the flight and entry phases.

A traditional approach to this performance problem would be to separate all the component parts into subcomponent parts and train them independently in both dry-land and pool training environments before “fitting” them together again in competition [44]. The practice design could include separate training sessions involving: upper body conditioning to help with shape coordination, lower body S&C work to improve the diver’s strength, power and balance, gymnastic-type acrobatic and somersault work on a AirTrack or tumble track, dry-diving board and trampoline (landing on foam mats) work on the approach and take-off phase and a mixture of pool skills that include standing and hurdle-step jumps and dives.

In contrast, a transdisciplinary ecological approach to facilitate improved balance on the diving board would seek to use both generality and specificity principles to enrich the interactions of the performer with the environment. Adopting a constraints-led approach, the diver is encouraged to search for and find an individualised solution. By strategically placing an object into the pool to guide intentions and provide a performance goal of “*land on or before the chamois*” [Figure 2], the diver will be encouraged to find a relevant, individualised solution to achieve the performance goal [Figure 3]. This practice task maintains the interconnectedness of the components of the dive and affords the diver the opportunity for greater action regulation. This task design offers rich opportunities for the performer to scale the complexity of the task (the diver can choose to place the object further out or closer in), therefore co-designing the practice environment to meet their individual needs [39].

Dry-land equipment such as trampolines can be useful for performers in more general practice tasks. Their performance functionality can be improved by learning how they can regulate their balance, power and height in the preparation phase of the dive. For example, using the rectangles and crash mats at the end of the trampoline affords the diver with minimal space to land in, or they will bounce off the trampoline. This task requires the diver to coordinate the trampoline action in such a way that they do not rotate forward onto the crash mat, potentially hurting themselves (Figure 4). Rather, the task constraints of the trampoline action guide them to regulate their functional capacities of power, balance and rotational speed to successfully navigate the task to land accurately and safely.

### 2.3. Enrichment in Team Sports (Rugby Union)

Age Grade Rugby [AGR] was introduced by the English Rugby Football Union [RFU] in 2016 to facilitate children’s (6–18 years) enjoyment in a safe environment, promoting holistic development and lifelong participation. Alongside AGR, the RFU also introduced seven Age Grade Rugby Codes of Practice (CoP) [47] as guidance for coaches in UK youth rugby union [47]. These range from minimum standards for coaches, grouping of players, individuals playing up and down age groups, out of season activities and keys to coaching children, developing the whole player and adopting a player-centred approach to playing and training [47,48]. To support coaches in building an individual-centred approach to develop the whole player, the RFU are using the *STEP Principle* as a coaching framework [49]. Coaches are encouraged to manipulate constraints based on space, task, equipment and people [STEP] within training and games to help structure sessions to promote holistic player development.

This case study outlines how a constraints-led approach can be used to meet the RFU CoP and to develop an individualised, ecological (child-centred) approach using the under 10 yrs (U10) AGR rules to enrich child-environment interactions in team sports. The U10 age grade rules allow for an eight-a-side game on a pitch size of 60 × 35 m. Games are played with a maximum of fifteen minutes per half [47], comprising tackling, a three-person uncontested scrum, and rucks and mauls which one support player from each team can join.

An example of a training game to afford children opportunities for running with the ball and finding space, as well as creating two attackers vs. one defender situations and a defensive team working together with less numbers, is *overload attack*. Under this type of task constraints, children can build physical condition to play rugby union, acquire relevant skills which can be adapted to different task constraints and learn to tactically interact with teammates and opponents in fundamental patterns. To represent the U10 game, this game should be designed on 60 × 35 m pitch with eight attackers against six defenders. If coaches have a bigger squad, they can design two games, one in each half of the pitch. If there is limited space, game design can have two defensive teams and one attacking team who attack a try line at each end against one defending team. Coaches can rotate the attacking team. Other *task constraints* in the game could challenge players to tackle, hold or touch. If there are new players involved, then a two-handed touch on the opponent’s waist could simulate a tackle. For any players lacking confidence in tackling at this stage, coaches could encourage a two-handed touch on the opponent’s waist to simulate a tackle. This task constraint allows those players to attune to the speed of the game, running lines of both attackers and defenders and also observe, at game speed, in-context tackling by other players, providing social affordances of others [45]. To form a ruck situation, once the ball carrier is held, tackled or touched, they must go to the floor and present the ball to their team. The tackler must also go to the ground if still standing and then get back on their feet. The closest attacking support player has 3 s to bear crawl past the tackler or their team lose possession of the ball. One defender must also bear crawl past the tackled player. These task constraints allow players to build strength and conditioning and encourage the adaptation of body height at the ruck. The next nearest attacker must then pass the ball away from the ruck area. This could be a designated player to include them in the game, or to develop passing. With two attackers and two defenders in each ruck this should leave a six vs. four overload attack. Scrums in the game, as with U10 rules, are uncontested but should involve the three nearest players from both attack and defence to form. Games should involve 5-min intervals and should be started with different scenarios, including: ball from a ruck in a specific area, a scrum in a position on the field, a pass in from the side of the field and rotating players. Using this format, coaches could take into consideration which players are competing with and against whom, which is so important to the dynamics of the game environment. Responsibility may be given to two attack and two defence team captains (to afford self-organisation and co-adaption) which can rotate after each 5-min interval. Those players may feed back their thoughts to the coach, teams and parents. Those captains could be allowed to theme their team, as superheroes, a pack of wolves or characters from films, which the children can decide.

As rugby union is an outdoor sport, environmental constraints are important to consider, for example, changing the direction of play if it is a windy day, or in bright sunshine, to offer different affordances for children. Playing times can be increased or decreased if it is very cold or hot, but play should only be 30 min in total. A key factor in building the learning environment is to involve parents, encouraging the parents for each child to hear and listen to the attack and defensive captains’ feedback after each interval in the game. Parents can be solicited to count number of successful, passes, runs and tackles of their child or a team, and the intended outcomes of the training can be explained to parents to promote understanding and encourage positivity. This type of ecological design should promote innovative and adaptive tactical behaviours within the practice environment and meet the RFU CoP requirements [11].

### 2.4. Generally Enriching S&C Designs for Sport Performance in Children and Youth

The primary goal of the S&C specialist in working with young athletes is to obtain changes in movement and physical capacities [22] that facilitate enhanced sports performance and injury prevention. In a typical “coach-centred” approach, this is attempted through explicit instruction that strives to align the young athlete’s technique with that which is considered “optimal” by the coach [50]. In contrast, an “individual-centred” approach involves less explicit instruction, having a reduced focus on a child executing each movement repetition in compliance with a putative “optimal” technique. An individual-centred approach provides opportunities for children to enhance their movement functionality by engaging in conditioning games, play activities and practice designs to explore diverse, relevant solutions to a movement problem [13].

Conditioning activities provide many opportunities to enrich youth-environment interactions. A broad range of physical attributes, such as speed, changing direction, agility and aerobic capacity may be simultaneously targeted and developed through participation in appropriately designed games to express movement skills [10]. A nuanced balance between generality and specificity of practice in children can also be used with adult athletes, who possess a higher training status and may require more specialised stimuli [i.e., focused sprinting activities] to continue to improve distinct physical qualities [51]. For example, with youth athletes, relay races involving actions with a ball, teammates and opponents and using playing area line markings, may be implemented to target acceleration, deceleration and change of direction or agility abilities, whilst handling the ball under pressure of competition, thereby maximising motivation, concentration and effort [10]. Similarly, small-sided games may be strategically designed to encourage athletes to cover greater distances (e.g., adapting their interactions with larger playing area dimensions, smaller player numbers, unequal sides, adding sudden opposition over-loads and teammate under-loads) providing more of a speed endurance stimulus [52]. The use of such interactive and dynamic games also provides an opportunity to integrate different aspects of performance development consistent with a transdisciplinary approach [i.e., psychological skills training, tactical adaptations, problem solving and decision making]. As an example, ensuring that children encounter feelings of uncertainty and lack of familiarity at different times in practice may provide them with opportunities to assert their ideas and express their autonomy, enhancing their communication and interaction skills. To illustrate, targeting specific instructions to individuals or sub-groups [e.g., attackers or defenders] not universally to all, could require children to work out what the opposition tactics are. Alternatively, deliberately providing unfair refereeing decisions may be embedded into skill games, without participant awareness, to promote the development of emotional self-regulation and coping skills [53].

Even in resistance-training activities (which involve less room for movement variability) information from external task constraints can be introduced to guide the learner towards effective and desirable movement solutions. For example, during squatting, jumping and landing movement patterns, resistance bands can be placed around the knees to encourage young athletes to perceive information from activating muscles necessary to overcome knee valgus. Similarly, during certain closed-chain kinetic exercises (i.e., push ups, inverted rows, etc.), promoting hip and trunk stiffness may help children maximise stability. This can be encouraged by placing a foam roller between the legs of the learner, who must contract the relevant hip and trunk musculature to keep the foam roller in position. As a final example, external objects (e.g., walls, agility poles, etc.) can be used to constrain dynamical movements and encourage the most biomechanically efficient and safe movement pattern for each young learner. By placing agility poles in front of a youth performing a power clean, the performer inherently learns to adopt a linear barbell path that remains close to the body [50]. Here, it is worth noting the similarities of using these external task constraints with the placement of the chamois in the case of helping divers to enhance spatial perception of diving into a specific pool location.

## 3. Conclusions

In this insights paper we have discussed why children and youth should not be treated as mini-adults by sport scientists in learning and development and preparation for performance. The expertise of professional specialists working in a department of methodology is important for design enrichment opportunities for children and youth in team games and individual sports. Our critical analysis showed that future research is needed for identifying in detail the needs and specific characteristics of specific sub-groups in sport, such as children and youth, women, and disabled athletes, at all levels, from recreation to high performance. Applied scientific research in various sport science sub-disciplines from the social and cultural sciences, bio-physical and pedagogical sciences, is needed to further identify and examine the needs of children and youth so that their performance trajectory and lived experiences in sport may promote enriched learning and training opportunities throughout that important part of the life course. The case examples showcased several types of designs which could fulfil this aim for children and youth in sport, framed by the contemporary ecological idea of enriching their interactions with performance and practice environments. These practice and training designs should facilitate a large amount of variability [54] providing many opportunities for play and games which facilitate implicit learning through interactions of children with sports environments. These practice designs for children and youth should emphasise a rich range of activities which integrate opportunities for skill adaptation, S&C and tactical and strategical behaviours. This important aim could be achieved by sport scientists collaborating together in a DoM to design affordances which invite functional actions for youth and children to adapt skills, perceive information to regulate actions, make decisions and solve problems, all while performing more or less dynamic movements at speed, showing endurance and using strength, flexibility and agility. Further research could be undertaken to show how professional practice, considered in this ecological way, could help collaborating sport scientists, trainers and coaches to *contextualise* learning designs and *individualise* the performance and development process in sports, based on the needs of individual children.

## Figures and Tables

**Figure 1 children-10-00752-f001:**
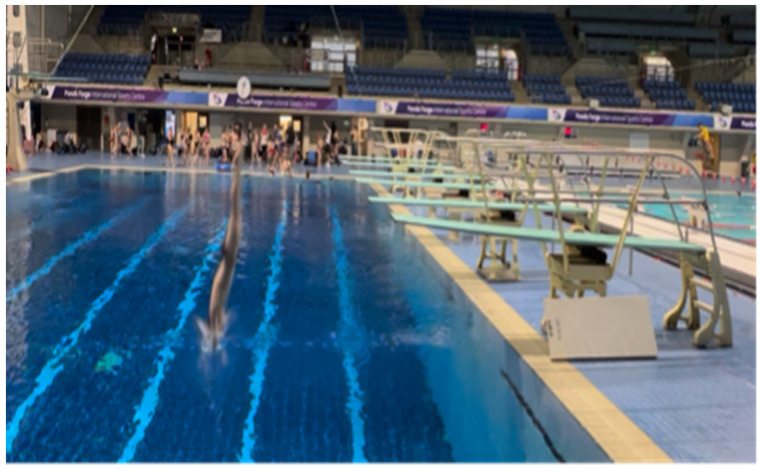
A snapshot of a diver landing too far away from the diving board.

**Figure 2 children-10-00752-f002:**
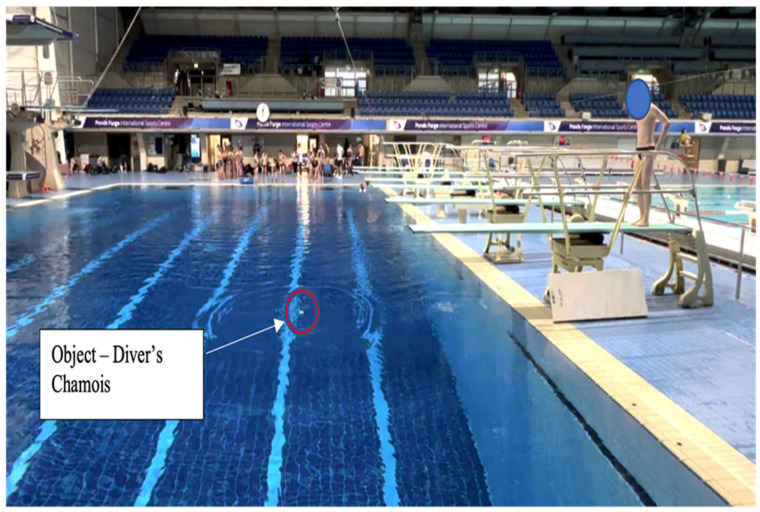
“Land on or before the chamois”: an example of a task constraint that could be used in the pool to help regulate where the diver lands. The positioning of the chamois cloth on the water surface specifies the location of the targeted entry position for the diver.

**Figure 3 children-10-00752-f003:**
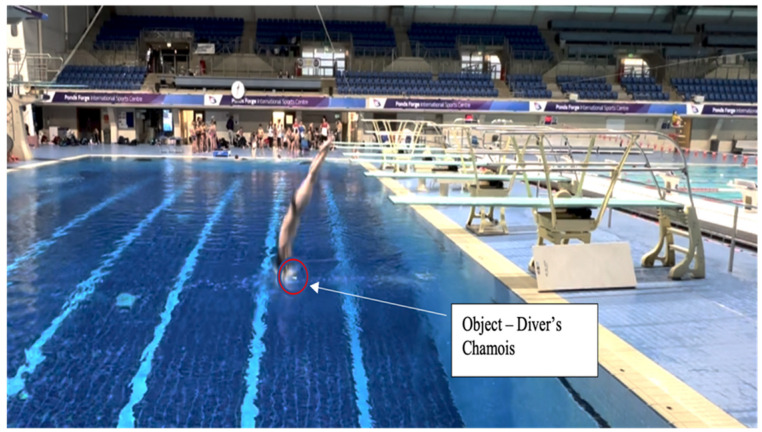
A snapshot of a diver’s landing and entry position as a result of the “land on or before the chamois” practice design.

**Figure 4 children-10-00752-f004:**
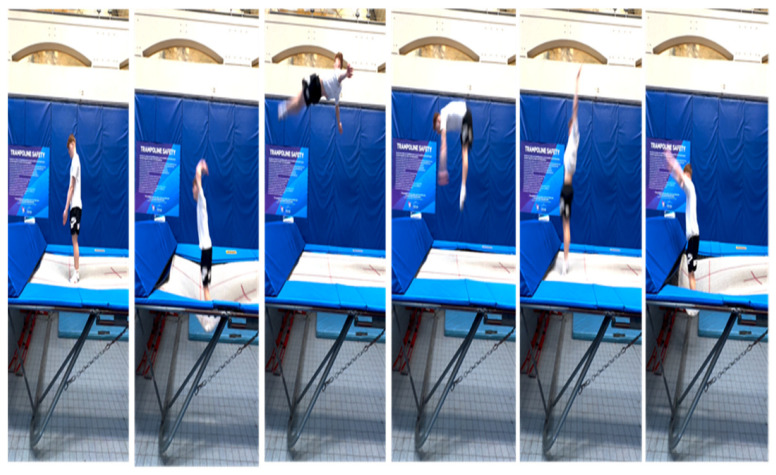
An example of a task constraint used on a trampoline in the dry-land practice area. The specific landing location on the trampoline that the diver is afforded guides them to regulate their functional capacities of power, balance and rotational speed to land accurately and safely.

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
