# Peer review of "Enriching Athlete—Environment Interactions in Youth Sport: The Role of a Department of Methodology"

_children, 2023, doi:10.3390/children10040752_

Round 1
Reviewer 1 Report
This impressive manuscript proposed to review how the theory of ecological dynamics may invite reconsideration of how sport scientists could support performance, learning and development of children and youth in sports programmes.
I think that the manuscript is very great for readers of children. However, few points must be modified prior the acceptance for publication.
-Introduction: to add the aim to this study.
-Methods: how were the papers revised? inclusion criteria?
-"Results/Discussion": to add info regarding to incentive of schools and parents to do exercise at school. In addition, what is opinion of educators and parents on food habits of children who do exercise and it importance to growth and development of children athletes.
Indded, this topic is important, more I suggest the authors to add the limitations of studies and to add a topic "new avenues to reaseacrhers in this area".
Author Response
The reviewer makes some very useful suggestions on the weaknesses of the submitted version of the manuscript which we have adopted to refine and improve the paper.
I think that the manuscript is very great for readers of children. However, few points must be modified prior the acceptance for publication.
-Introduction: to add the aim to this study.
Authors Response: This is an important addition to the paper. We have outlined the aim in the abstract for readers and we have then elaborated on the aims between lines 55 to 65.
-Methods: how were the papers revised? inclusion criteria?
Authors Response: Again this is an important addition and we added material on lines 66-73 to explain our review strategy.
-"Results/Discussion": to add info regarding to incentive of schools and parents to do exercise at school. In addition, what is opinion of educators and parents on food habits of children who do exercise and it importance to growth and development of children athletes.
Authors response: Excellent idea. We related this suggestion to Bronfenbrenner's ecological model, especially focused on the meso level of analysis (see lines 330-338).
Indded, this topic is important, more I suggest the authors to add the limitations of studies and to add a topic "new avenues to reaseacrhers in this area".
Reviewer 2 Report
The abstract does not explain the objective, methodology and study design.
The first paragraph should be deleted or shortened as it does not have much to do with the research, a brief overview of it would be correct, such a long paragraph has no relation to the object of the study. (15-31)
lines 47-55. Authors who refer to this difference between children and adults in training should be cited, as there are many quotes that talk about the different methodologies to be used in sports science, and they do not cite anything about it.
Basically in this first part they are based on two authors, it is true that they support them with other works, but this part should be expanded and improved between lines 14 and 61.
Between lines 123 and 140 there is no citation and therefore this is inadequate for a quality scientific article. Just as the use of technology is well researched and is of great help to the understanding of young players as has already been shown, they should cite these studies that talk about it today.
Again, between lines 156-169, current studies dealing with decision making, ecological vision... should be cited.
between lines 170-228 there are again no citations on the contents that are argued, when there are quite a few studies that already analyse what you indicate.
between lines 256-284 there are again no citations on the contents that are argued, when there are quite a few studies that already analyse what you indicate.
Line2 285-322 In this section there are many studies of various sports that support the indications provided by the authors but they are not cited and should cite more studies on this subject, especially because depending on the element to be worked and the age the types of training may be different and that would support the ideas expressed by the authors.
Again, between lines 323-343, no research studies are cited that analyse similar issues and would support the ideas expressed by the authors.
For example (392-393): When are SSGs needed over full sided games for example? Or when might an individual sport athlete/child need to compete up or down an age group? What specific effects may specific SSGs have on children’s effectivities (e.g., 2v2, 394 4v4 and 7v7)? = This is already studied and scientifically proven and therefore it would be necessary to explain why they indicate this as a question or hypothesis if the literature has already described the varied effect of this type of training in young athletes?
Lines 396-425: Again, there is a lack of research to support these ideas or reflections, which should be cited in order to give them greater scientific rigour.
In the article they focus on a methodology department but do not focus on the use of training and teaching methodologies that have already made great strides in this field. Furthermore they do not explain the various methodological uses nor do they focus on the use of methodology, traditional and non-linear or alternative, they focus on the study and analysis of the effect it has on athletes. They also do not cite current studies that support their ideas and reflections, which in some cases, are proven in one sense or another.
Author Response
The reviewer made some useful suggestions which we have responded to below:
The abstract does not explain the objective, methodology and study design.
Authors Response: We have clarified on line 51 that this paper is a critical analysis which reviews literature relevant to an important issue for sport scientists, especially those working with children and youth. We explained that our aim was to review the literature and set up a critical analysis, focusing on ecological dynamics’ literature.
The first paragraph should be deleted or shortened as it does not have much to do with the research, a brief overview of it would be correct, such a long paragraph has no relation to the object of the study. (15-31)
Authors Response: The first paragraph has been shortened in line with the reviewer comments, with some material deleted. It is important to note that the ‘adultification’ of children’s experiences in sport can be traced back in history, explaining that this feature of human life has been around for some time.
lines 47-55. Authors who refer to this difference between children and adults in training should be cited, as there are many quotes that talk about the different methodologies to be used in sports science, and they do not cite anything about it.
Authors Response: We have now cited many references on differences between children and adults and explained that they have summarised data from studies (starting on line 50). The current submission seeks to contribute discussion on sport science support for enriching interactions of children and youth with practice environments, from literature on ecological dynamics.
Basically in this first part they are based on two authors, it is true that they support them with other works, but this part should be expanded and improved between lines 14 and 61.
Authors Response: Some relevant exemplar references that have examined individual-environment interactions (not either separately) have been added to the text (see references now included between lines 50 to 80). To achieve this specific requirement to amendments to citations, we added some exemplar references to address the next 6 points below.
Between lines 123 and 140 there is no citation and therefore this is inadequate for a quality scientific article. Just as the use of technology is well researched and is of great help to the understanding of young players as has already been shown, they should cite these studies that talk about it today.
Again, between lines 156-169, current studies dealing with decision making, ecological vision... should be cited.
between lines 170-228 there are again no citations on the contents that are argued, when there are quite a few studies that already analyse what you indicate.
between lines 256-284 there are again no citations on the contents that are argued, when there are quite a few studies that already analyse what you indicate.
Line2 285-322 In this section there are many studies of various sports that support the indications provided by the authors but they are not cited and should cite more studies on this subject, especially because depending on the element to be worked and the age the types of training may be different and that would support the ideas expressed by the authors.
Again, between lines 323-343, no research studies are cited that analyse similar issues and would support the ideas expressed by the authors.
For example (392-393): When are SSGs needed over full sided games for example? Or when might an individual sport athlete/child need to compete up or down an age group? What specific effects may specific SSGs have on children’s effectivities (e.g., 2v2, 394 4v4 and 7v7)? = This is already studied and scientifically proven and therefore it would be necessary to explain why they indicate this as a question or hypothesis if the literature has already described the varied effect of this type of training in young athletes?
Authors Response: The current submission is not a report on an experimental study and therefore does not require a hypothesis. Rather, as we have clarified in the abstract and introduction now it is intended as a critical analysis and related review of the literature which specifically addresses the enrichment of child-environment interactions, at an ecological scale of analysis. Some of the questions raised in this point seem to imply that the analysis is of studies adopting an environmental or a person-oriented perspective separately. These perspectives on human behaviours, generally, are not considered ecological in orientation as James Gibson (1979) argued.
Lines 396-425: Again, there is a lack of research to support these ideas or reflections, which should be cited in order to give them greater scientific rigour.
In the article they focus on a methodology department but do not focus on the use of training and teaching methodologies that have already made great strides in this field. Furthermore they do not explain the various methodological uses nor do they focus on the use of methodology, traditional and non-linear or alternative, they focus on the study and analysis of the effect it has on athletes. They also do not cite current studies that support their ideas and reflections, which in some cases, are proven in one sense or another.
Authors response: While there have been relevant contributions made by previous publications from more traditional viewpoints, our aim in this paper was to provide a critical review of relevant papers written and published from an ecological perspective (focusing on enrichment processes for children and youth, with the appropriate person-environment scale of analysis as advocated by Gibson (1979). This literature base has proposed adoption of a transdisciplinary perspective, implying the need for a collaborative approach by sport scientists working in a Department of Methodology, underpinned by ecological principles. This paper summarises and reviews key ideas that have been argued in previous theoretical, applied scientific rationales and practical applications in an ecological perspective.
Reviewer 3 Report
Many thanks for the opportunity to contribute to the pre-publication peer review process for the original perspectives submission to Children titled “Enriching athlete-environment interactions in youth sport: The role of a Department of Methodology” (children-2290704).
The following comments and suggestions are offered to the authors to assist the evolving of the reporting of their 'perspective':
Accepting this submission is under the 'Perspectives' category, it is still recommended that the authors consider depersonalizing the manuscript. For example, Page 1, Abstract, line 4 '...paper, it is proposed how...'; line 6 '...programmes. This outline...'; Page 2, Introduction, line 55 '...paper it is highlighted how...', and so on.
Page 4, line 156 - the in text citation here provides a page number, but there is seemingly no obvious direct quotation evident within text - it is suggested that the authors italicize or use quotation marks to identify to reader direct sentiments ascribed to the citation.
Across the submission there are numerous words or phrases italicized, but not as ascribed to a citation; is there a rationale for this - is it for effect/emphasis? Is this consistent with the style guide (this reviewer will defer to the editorial office team on this)?
Page 7, lines 311-2 - the same Gibson (1979) reference is cited twice within same sentence - suggest review such that citation given singularly at end of sentence.
line 337 - abbreviation 'S&C' used here without prior defining. It is also noted that 'strength and conditioning' subsequently appears twice (line 381 & 574) - suggest review for defining and consistency of use of abbreviation.
Line 392 - abbreviation 'SSGs' used here without prior defining - only appears once more on line 394, so suggest just write out in full both times.
Line 399 - abbreviation 'LTAD' defined here, but then not used again. Same for 'YPD' on line 400 - suggest review for utility.
Line 468 - suggest review spacing/typesetting
References - suggest review capitalization of article titles for consistency and alignment with style guide.
Similarly suggest review capitalization of journal titles as these too are inconsistent within list.
Lines 691-2 - review italicizing of journal title
Inclusion and format of doi is also inconsistent - suggest review
Some references have the volume italicized, most don't. Some references include an issue in parentheses, most don't - suggest that these minor variations and inconsistencies in style be reviewed/remediated.
Page 18 - correct label as Figure 1
Author Response
The reviewer helpfully picked up some errors and issues in the first version of the manuscript. We have responded to these suggestions below:
Many thanks for the opportunity to contribute to the pre-publication peer review process for the original perspectives submission to Children titled “Enriching athlete-environment interactions in youth sport: The role of a Department of Methodology” (children-2290704).
The following comments and suggestions are offered to the authors to assist the evolving of the reporting of their 'perspective':
Accepting this submission is under the 'Perspectives' category, it is still recommended that the authors consider depersonalizing the manuscript. For example, Page 1, Abstract, line 4 '...paper, it is proposed how...'; line 6 '...programmes. This outline...'; Page 2, Introduction, line 55 '...paper it is highlighted how...', and so on.
Authors Response: There have been some changes to the text to make the writing less oriented towards a personal viewpoint, adopting a more passive writing style in many places. See changes to Abstract and Introduction.
Page 4, line 156 - the in text citation here provides a page number, but there is seemingly no obvious direct quotation evident within text - it is suggested that the authors italicize or use quotation marks to identify to reader direct sentiments ascribed to the citation.
Authors Response: Thanks for picking this error up. Page number not needed and now deleted.
Across the submission there are numerous words or phrases italicized, but not as ascribed to a citation; is there a rationale for this - is it for effect/emphasis? Is this consistent with the style guide (this reviewer will defer to the editorial office team on this)?
Author Response; You are right, the italicisation of certain words was intended by the authors. This formatting style is often used as a literary device to draw the readers’ attention to key words and messages in a book, chapter or paper. Does this work? Perhaps, as your attention was drawn to these words. If editorial staff of the journal do not wish to include such italicisation in the text, they will change this formatting at the proof reading stage.
Page 7, lines 311-2 - the same Gibson (1979) reference is cited twice within same sentence - suggest review such that citation given singularly at end of sentence.
Authors Response: A good idea to delete the Gibson reference to the concept of affordances in that sentence. Whilst the reference is accurate, it is not needed since the concept of affordances was referenced earlier in the submission already.
line 337 - abbreviation 'S&C' used here without prior defining. It is also noted that 'strength and conditioning' subsequently appears twice (line 381 & 574) - suggest review for defining and consistency of use of abbreviation.
Authors Response: We have amended this writing style and the next 2 suggestions in line with the reviewer’s comments
Line 392 - abbreviation 'SSGs' used here without prior defining - only appears once more on line 394, so suggest just write out in full both times.
Line 399 - abbreviation 'LTAD' defined here, but then not used again. Same for 'YPD' on line 400 - suggest review for utility.
Line 468 - suggest review spacing/typesetting
Authors Response: All amended
References - suggest review capitalization of article titles for consistency and alignment with style guide.
Similarly suggest review capitalization of journal titles as these too are inconsistent within list.
Lines 691-2 - review italicizing of journal title
Inclusion and format of doi is also inconsistent - suggest review
Some references have the volume italicized, most don't. Some references include an issue in parentheses, most don't - suggest that these minor variations and inconsistencies in style be reviewed/remediated.
Page 18 - correct label as Figure 1
Author Response: Corrected as recommended
Round 2
Reviewer 2 Report
All correct answers.